# Supraphysiological Oxygen Levels in Mammalian Cell Culture: Current State and Future Perspectives

**DOI:** 10.3390/cells11193123

**Published:** 2022-10-04

**Authors:** Ricardo Alva, Georgina L. Gardner, Ping Liang, Jeffrey A. Stuart

**Affiliations:** Department of Biological Sciences, Brock University, St. Catharines, ON L2S 3A1, Canada

**Keywords:** oxygen, physioxia, hyperoxia, ROS, oxidative stress, gene expression, senescence, metabolism, mitochondrial dynamics, drug response

## Abstract

Most conventional incubators used in cell culture do not regulate O_2_ levels, making the headspace O_2_ concentration ~18%. In contrast, most human tissues are exposed to 2–6% O_2_ (physioxia) in vivo. Accumulating evidence has shown that such hyperoxic conditions in standard cell culture practices affect a variety of biological processes. In this review, we discuss how supraphysiological O_2_ levels affect reactive oxygen species (ROS) metabolism and redox homeostasis, gene expression, replicative lifespan, cellular respiration, and mitochondrial dynamics. Furthermore, we present evidence demonstrating how hyperoxic cell culture conditions fail to recapitulate the physiological and pathological behavior of tissues in vivo, including cases of how O_2_ alters the cellular response to drugs, hormones, and toxicants. We conclude that maintaining physioxia in cell culture is imperative in order to better replicate in vivo-like tissue physiology and pathology, and to avoid artifacts in research involving cell culture.

## 1. Introduction

At sea level, dry atmosphere is composed of ~21% O_2_. Incubators used in standard cell culture practices regulate headspace CO_2_ and temperature, but the vast majority do not actively regulate O_2_. Although this is usually referred to in the literature as 20% or 21% O_2_, in fact, the headspace O_2_ in a humid cell culture incubator maintained at 5% CO_2_ typically equilibrates to ~18% in the absence of O_2_ regulation. The overwhelming majority of in vitro studies have been performed in these conditions, and, as such, 18–21% O_2_ is commonly termed ‘normoxia’ in the literature [1]. While atmospheric O_2_ may be a normoxic condition for an entire organism, the same is not true for mammalian cells in vivo. O_2_ levels in most mammalian tissues range from 2–6% [2], i.e., three- to nine-times lower than standard cell culture normoxia. In this review, we will use the term ‘physioxia’ [3] to describe O_2_ levels in this 2–6% range, and ‘normoxia’ for the hyperoxic (18%) conditions of standard cell culture. We are, thus, interested here in the effects of O_2_ in the range between physioxia and normoxia, that is, the ‘physio-normoxic’ range. It is important to note that, although most authors refer to O_2_ levels in experiments where cells are grown in humidified cell culture incubators with 5% CO_2_ as being either 20% or 21% O_2_, these numbers are incorrect. We have used the more accurate 18% throughout this text, though we recognize that this too will depend on elevation above sea level.

The appreciation that standard cell culture conditions are in fact hyperoxic, compared to what mammalian cells encounter in vivo has led to efforts to replicate physiological O_2_ conditions (physioxia) in vitro. In this review, we present our current understanding of how cellular processes are affected by O_2_ in the range from physioxia to normoxia, including reactive oxygen species (ROS) generation and clearance, gene expression, proliferation, replicative lifespan, bioenergetics, and mitochondrial function. In addition, we discuss how cells cultured in normoxia exhibit phenotypes that fall short of reproducing tissue physiology and pathology in vivo. In this light, we consider recent findings showing that cells respond to drugs, hormones, and toxicants differently when cultured in physioxia, compared to 18% O_2_. We finalize by commenting on solutions to these challenges and future research directions.

## 2. Oxygen and Redox Homeostasis

The different types of ROS, their generation, and their clearance have been extensively investigated and reviewed elsewhere [4,5,6]. Thus, we will mainly focus on how ROS metabolism is affected in the physio-normoxic range.

Superoxide anion (O_2_^−^) is formed by single electron reduction of O_2_. O_2_^−^ is rapidly dismuted into hydrogen peroxide (H_2_O_2_) by superoxide dismutases (SOD) 1–3 [7]. Due to its neutral charge and diffusivity, H_2_O_2_ is able to cross membranes and participate in signaling processes by oxidizing cysteine residues in key signaling proteins that act as redox-sensitive molecular switches [8]. H_2_O_2_ is transformed into water by catalase, glutathione peroxidase (GPx), thioredoxin (Txn), and peroxiredoxin (Prdx) [9]. In the presence of Fe^2+^, H_2_O_2_ is converted into the highly reactive hydroxyl radical (HO·), which indiscriminately oxidizes lipids, proteins, and DNA [10]. Although ROS have well-characterized roles in cellular physiology (reviewed in [11]), overproduction of ROS can be detrimental, which is why it is important to study ROS production rates from different sources and understand the factors that regulate each.

There are multiple sources of ROS in mammalian cells, with the most studied being mitochondrial enzymes (e.g., respiratory complexes I and III) and NADPH oxidases (NOX). The *K_M_* values of the complex I flavin mononucleotide (FMN) site and complex III Qo site toward O_2_ are 0.019% and 0.199%, respectively [12], meaning that in the 2–18% O_2_ range, these enzymes are likely saturated and that little difference would thus be expected in mitochondrial ROS (mROS) production of cells grown at 18% O_2_ versus physioxia. Nonetheless, some studies have reported higher ROS levels at 18% O_2_, using mitochondria-targeted ROS indicators such as MitoSOX and dihydrorhodamine 123 [13,14,15], suggesting increased mROS production at normoxia (18% O_2_). On the other hand, NOX isoforms have higher *K_M_* values. For example, NOX4 has a reported *K_M_* of ~18% O_2_ [16], indicating that the reaction rate is highly sensitive to O_2_ concentration in the 2–18% range. Accordingly, production of H_2_O_2_ in C2C12, PC-3, SH-SY5Y, HeLa, MCF-7, and mouse embryonic fibroblasts (MEFs) was found to be significantly higher at 18% O_2_ than at 5% O_2_ [17]. Pharmacological inhibition of NOX1/4 abrogated this effect, confirming the importance of NOX enzymes in ROS production at O_2_ levels higher than physiological conditions. A thorough discussion of how O_2_-consuming reactions are impacted by O_2_ tension in cell culture is provided in [18]. Indeed, elevated ROS production at supraphysiological O_2_ levels has been demonstrated by numerous studies in multiple cell types [13,14,15,19,20,21]. In addition, significantly lower reduced glutathione (GSH) levels have been observed in cells grown at 18% O_2_ [20,22,23]. Thus, there is ample evidence that ROS production is greater in normoxia, compared to in physioxia, often compromising the antioxidant defenses of cells.

Disruption of the balance between ROS production and antioxidant defenses results in oxidative stress, characterized by damage to lipids, proteins, and DNA. Timpano et al. reported that U87MG, HRPTEC, HEK293, and MCF-7 cells grown at 18% O_2_ exhibit more DNA strand breaks than their counterparts cultured in physioxia, as determined by comet assay [24]. Further, increased levels of 8-oxodeoxyguanine and 8-oxoguanine were observed in all cell lines but MCF-7, indicating oxidative damage to the nitrogen bases of DNA and RNA in these cells grown at 18% O_2_. Protein oxidation was observed in rat myoblast precursor cells (MPCs). Protein carbonyl levels, one of the most common biomarkers of protein oxidation, were higher at 18% O_2_, compared to 3% O_2_, in old MPCs only, with no effects observed in cells from young mice [25]. Human mesenchymal stem cells (hMSCs) also showed increased protein carbonyl content and DNA strand breaks when cultured in ambient air [19]. In addition, hMSCs grown at 18% O_2_ had higher levels of malonyl aldehyde (MDA), which is an end-product and biomarker of lipid peroxidation. Elevated MDA levels were also detected in human dental pulp stem cells (hDPSCs) grown at 18% O_2_ for 7 days, compared to cells grown at 3% O_2_, which was prevented with Trolox administration, a water-soluble vitamin E analog [15]. Finally, levels of both protein carbonyls and 4-hydroxynonenal (4-HNE), another lipid peroxidation end-product, were found to be higher in mouse cardiac fibroblasts cultured at 21% O_2_, compared to 3% O_2_ [13].

To conclude, there is abundant evidence that the hyperoxic environment of standard cell culture disrupts redox homeostasis by increasing the production of ROS, which results in oxidative damage to biomolecules and organelles. Biomolecule oxidation can, in turn, result in the dysregulation of signaling pathways and cellular processes, such as gene expression, proliferation, senescence, differentiation, metabolism, and response to drugs and other bioactive molecules.

## 3. Intracellular Oxygen and ROS Sensing

Discovered in the early 1990s, the hypoxia-inducible factor (HIF) is regarded as the master intracellular O_2_ sensor in response to hypoxia [26,27]. HIF is a heterodimer composed of one α and one β subunit. While there are three isoforms of HIF-α (HIF-1α, HIF-2α, and HIF-3α, encoded by the genes *HIF1A*, *HIF2A*, and *HIF3A*, respectively), only one HIF-β protein has been described (reviewed in [28]). When translocated to the nucleus, HIF-α partners with the constitutively expressed HIF-β, binds to the hypoxia response element (HRE) of DNA and initiates transcription of target genes. The transcriptional activation of HIF allows the cell to adapt to low O_2_ tensions, by mediating expression of genes involved in glycolysis and energy metabolism, angiogenesis, cell migration, proliferation, and survival, among other processes [29]. Although HIF-1α and HIF-2α have unique gene targets, both are critical for regulating the expression of genes necessary for cell adaptation to hypoxia [30] (reviewed in [31]). In normoxia, prolyl hydroxylases (PHD) use molecular O_2_ as substrate to introduce hydroxyl groups in two highly conserved prolyl residues of HIF-α. Hydroxylated HIF-α is then recognized by the Von Hippel-Lindau tumor suppressor protein (pVHL), which leads to its ubiquitination and subsequent degradation by the proteosome [32]. In contrast, a low O_2_ concentration in hypoxia blunts PHD activity and allows for the stabilization of HIF-α and its translocation to the nucleus (Figure 1A) [33]. It is important to note that nearly all studies on HIF-1/2 activity have been conducted as comparisons between normoxia (18%) and hypoxia (≤1% O_2_). Future studies should be conducted comparing physioxia and hypoxia.

While HIF activation has long been considered solely as a cellular response to hypoxia, a few studies have also suggested its role in physiological O_2_ conditions. Given that the *K_M_* values of PHD isoforms are slightly above atmospheric O_2_ level [34], the enzymatic rate of HIF-α hydroxylation is highly sensitive to O_2_ levels in the physio-normoxic range. Bracken et al. investigated the stabilization of HIF-1/2α in six cell lines, HeLa, 293T, Cos-1, PC-12, CACO2, and HepG2, and they observed cell-type specific patterns [35]. For example, HIF-1/2α were barely detectable in HeLa cells at 5% O_2_ but upregulated at 2% O_2_. On the other hand, 293T cells had higher HIF-1/2α levels at 5% O_2_. These results likely reflect different sensitivities of distinct cell types to O_2_ levels [35]. In another study, Yan et al. detected HIF-1α localization to the nuclei of primary hepatocytes at 5% O_2_ via immunofluorescence [36]. More recently, detectable protein levels of HIF-1α and HIF-2α were observed in patient-derived melanoma cells cultured at 6% O_2_, supporting the notion that HIF is partially active at physiologically relevant O_2_ tensions [37]. Moreover, in this study, increased expression of HIF targets was sustained in cells grown at 6% O_2_ for at least 3 weeks, further suggesting a role for HIF activity at physioxia rather than it being a mere consequence of an acute reduction of O_2_ tension while performing the experiments. Stuart et al. have hypothesized that transcriptional activity of HIF-1/2 at low O_2_ tensions within a physiological range (2–6%) helps to maintain the production of ROS and reactive nitrogen species (RNS), which are necessary for many physiological processes [38]. This hypothesis is based on the fact that several ROS-producing enzymes are induced by HIF-1/2 (reviewed in [18]).

In contrast to direct oxygen-sensing mechanisms in hypoxia, the cellular response to hyperoxia is regulated by molecular switches that are sensitive to H_2_O_2_. Perhaps the most important and best described is the pathway regulated by the nuclear factor erythroid 2–related factor 2 (Nrf2). The molecular mechanisms regulating Nrf2 degradation, stabilization, and transcriptional activity have been thoroughly reviewed in [39,40,41]. Under unstressed conditions, Nrf2 is bound to Kelch-like ECH-associated protein 1 (Keap1), which facilitates ubiquitination and proteasomal degradation of Nrf2. Keap1 acts as an intracellular ROS sensor, with multiple cysteine residues identified as redox-sensitive regulators. In conditions of oxidative stress, such as hyperoxia, ROS-mediated oxidation of specific cysteine residues inhibits Keap1 binding to Nrf2, leading to Nrf2 stabilization, translocation to the nucleus, and binding to the antioxidant response element (ARE) of DNA (Figure 1B) [42]. In addition to oxidants, the regulatory cysteine residues of Keap1 are also sensitive to electrophiles that readily react with thiol groups [40]. Such electrophiles can originate from both exogenous and endogenous sources. In the latter case, for instance, the lipid peroxidation end-product 4-HNE reacts with specific cysteine residues of Keap1, promoting Nrf2 stabilization and transcriptional activity [43,44]. Nrf2 induces the expression of genes involved in ROS and xenobiotic detoxification, GSH biosynthesis and regeneration, NADPH regeneration, and heme and iron metabolism [45].

It is important to note that Nrf2 is part of the cellular response to oxidative and electrophilic stresses, which may be caused by a variety of insults, not just hyperoxia. Nevertheless, hyperoxia is a well-characterized condition of increased ROS production, in which the role of Nrf2 has been demonstrated. Indeed, activation of Nrf2 in severe hyperoxia (>60% O_2_) has been extensively studied (reviewed in [46]). Its role in perceived hyperoxia at ambient air (~18% O_2_) in vitro, on the other hand, has been less studied. However, a handful of studies have reported upregulation of Nrf2 in cells cultured in normoxia, compared to cells grown in physioxia. Epithelial cells from mouse mammary tumors processed under ambient air had increased levels of Nrf2 and decreased levels of Keap1, compared to tumors under physioxia (3% O_2_) [47]. In accordance, upregulation of known Nrf2 targets, such as SOD1, thioredoxin reductase, and NAD(P)H quinone dehydrogenase (NQO1), has been observed in normoxia versus physioxia in different cell types [15,23,24]. Due to the induction of antioxidant and cytoprotective enzymes in normoxia (~18% O_2_), some studies have shown that cells adapted to these conditions may be preconditioned and, thus, resistant to pro-oxidant insults (e.g., drugs and toxins). This shall be further addressed in Section 9. Moreover, mechanisms of enhanced Nrf2-dependent gene transcription in normoxia may go beyond increased Keap1 oxidation or downregulation. Chapple et al. demonstrated nuclear accumulation and DNA binding of Nrf2 in human endothelial cells grown in physioxia [48]. Instead, Bach1, another Nrf2 repressor, was found to be upregulated in physioxia, thus interfering with expression of Nrf2 targets. Inhibition of Nrf2 transcriptional activity was abrogated by re-exposure to ambient air or Bach1 silencing. This provides a novel mechanism for Nrf2-mediated regulation of gene expression induced by O_2_ in the physio-normoxic range.

Other ROS-activated pathways known to be affected in hyperoxia include those mediated by the nuclear factor kappa B (NF-κB) and mitogen activated protein kinases (MAPKs) (reviewed in [46]), but their regulation in physioxia versus atmospheric O_2_ has been scarcely investigated. Carrera et al. did observe increased phosphorylation of extracellular signal-regulated protein kinase (ERK1/2), in HCT116 and U2OS cells grown at 5% O_2_ in the absence of any stress [49]. They showed that the constitutive activation of this pro-survival pathway affects their response to proapoptotic drugs (see Section 9). Indeed, differential regulation of O_2_-sensitive and redox-sensitive signaling molecules and transcription factors leads to the alteration of multiple biological processes, as shall be further discussed throughout this review. Additional research is required to fully elucidate the roles and molecular mechanisms of HIF, Nrf2, and other pathways in the cellular adaptation to supraphysiological O_2_ in different cell types.

## 4. Effects of Oxygen and ROS on Gene Expression

Differential regulation of O_2_/ROS-sensitive transcription factors in the physio-normoxic range affects gene expression, as we have recently shown in an RNA-seq-based study with LNCaP, Huh-7, PC-3, and SH-SY5Y cells. These cell lines were cultured at either 5% or 18% O_2_ for two weeks, prior to mRNA isolation and sequencing. The O_2_ level significantly affected the abundance of thousands of transcripts in all cell lines. Surprisingly, there was limited overlap in the specific transcripts affected, with >80% of the differentially expressed genes being cell-line specific [50]. Interestingly, a handful of HIF targets were found to be upregulated at 5% O_2_ in LNCaP and Huh-7 cells, further confirming a role for HIF activation under physiological O_2_ conditions. Similar patterns were found by Duś-Szachniewicz et al. at the proteome level [51]. They analyzed proteomic changes in three diffuse large B-cell lymphoma cell lines grown at 1% O_2_ (hypoxia), 5% O_2_ (physioxia), and 18% O_2_ (normoxia). Importantly, proteomes from cells grown in hypoxia and physioxia cells shared more similarities, compared to the proteome of cells grown in atmospheric O_2_. In agreement with our study, this latter study showed that the effects of physioxia versus normoxia were largely cell-type specific, even when the three cell lines are from the same type of cancer [51].

Mechanisms of translation are also regulated by O_2_ (reviewed in [52,53]). Hypoxia prevents the binding of the eukaryotic translation initiation factor 4E (eIF4E) to the 7-methylguanosine cap at the 5′ end of mRNA molecules, which is the initial step of protein synthesis. In 2012, Uniacke et al. demonstrated that, under hypoxic conditions, HIF-2α forms a complex with eIF4E2 (eIF4E homologue) and the RNA binding motif protein 4 (RBM4), which then recognizes and binds to the 5′ cap of specific mRNA molecules, allowing initiation of translation independent of eIF4E [54]. Years later, the same research group found that, while eIF4E is the dominant cap-binding protein in normoxia (18% O_2_) and eIF4E2 is the dominant cap-binding protein in hypoxia, both are active at physioxia (here, 3–12%) and simultaneously act to initiate the translation of distinct classes of mRNA molecules [55]. These results demonstrate that differential expression of proteins in physioxia versus normoxia is not only regulated at the transcriptional level but also at the translational level.

In addition to transcriptional and translational regulation, O_2_-dependent mechanisms mediate gene expression through epigenetic modifications. Processes such as histone methylation and acetylation, DNA methylation, and chromatin organization are known to be affected by hypoxia and oxidative stress (reviewed in [56,57]). The roles of these mechanisms in the physio-normoxic range have been investigated less often, although a handful of studies have reported different epigenetic responses to O_2_ in a variety of cells, with a special focus on stem cells and cancer cells.

Lengner et al. found that culturing human embryonic stem cells (hESCs) in physioxia prevents chromosome X inactivation that is otherwise observed in normoxia, which, in turn, preserves the pluripotency of hESCs [58]. This was associated with the methylation of the *XIST* promoter region, while chromosome X inactivation was associated with the demethylation and transcriptional activity of *XIST*. In another study, Xie et al. observed an increased methylation pattern and irreversible silencing of the *DLK1-DIO3* gene cluster in cultured hESCs after 20 passages in normoxia [59]. Interestingly, culture of hESCs in 5% O_2_ preserved expression of the *DLK1-DIO3* cluster. The *DLK1-DIO3* cluster has been associated with pluripotency in mouse induced pluripotent stem cells and may, thus, be a biomarker for the epigenetic stability of hESCs. Further, acetylation of histone protein H3 at lysine residues 9 and 27 (H3K9ac and H3K27ac, respectively) was elevated in human pluripotent stem cells (hPSCs) grown at 5% O_2_, while trimethylation of H3 lysine 27 was reduced [60]. These epigenetic changes are associated with a more open chromatin structure and enhanced gene expression. RNA-seq and functional enrichment analysis also revealed increased expression of genes involved in H3K27 demethylation at 5% O_2_, while increased methyltransferase and cell cycle activities were observed at 18% O_2_.

The ten-eleven translocation (TET) family of proteins catalyzes the O_2_-dependent hydroxylation of 5-methylcytosine (5mC) in DNA, which results in demethylation and concomitant increase in gene expression. Thienpont et al. investigated the effect of tumor hypoxia (0.5% O_2_) on the expression profiling and activities of TET isoforms, TET1, TET2, and TET3 [61]. The mRNA levels of *TET* paralogues were increased at a variable degree among a range of cell lines exposed to hypoxia, with neuroblastoma cell lines SH-SY5Y and SK-N-Be2c being the most strongly affected. Chromatin immune precipitation followed by sequencing (ChIP–seq) confirmed the binding of HIF near the promoters of upregulated *TET* genes. However, the activity of TET enzymes was actually reduced in hypoxic human and mouse tumor cells, which is not surprising, since these enzymes use O_2_ as substrate. Reduced TET activity was then correlated with decreased hydroxylated 5mC (5hmC) and concomitant hypermethylation. Although these landmark results provided a mechanism for epigenetic-mediated repression of tumor suppressor genes in hypoxia, no changes in TET activity were observed at 2–5% O_2_. In a different study, however, the expression of the three *TET* paralogues was increased in HepG2 cells at both 5% and 1% O_2_, compared to 18% O_2_ [62]. Upregulation of these enzymes at 1% O_2_ was blunted by HIF-1α knockdown, although this was not tested at 5% O_2_. The activity of TET enzymes was not measured in this study. In stem cells, on the other hand, a different trend in the regulation of *TET* genes has been observed. *TET1* mRNA and protein levels were decreased in bone-marrow-derived hMSCs grown at 2% O_2_, compared to atmospheric O_2_ [63]. Expression of DNA methyltransferase 3B (*DNMT3B*) was also reduced at 2% O_2_. As a result, global 5hmC and 5mC levels were also decreased at physioxia. In a subsequent study by the same lab, *DNMT3B*, *DNM3L*, *TET1*, and *TET3* were similarly downregulated in hPSCs cultured at 2% O_2_, compared to cells grown at 18% O_2_ [64]. This was accompanied by enhanced proliferation, metabolic activity, and stemness attributes. Interestingly, both studies showed downregulation of *HIF1A* and upregulation of *HIF2A* (in hMSCs and hPSCs, respectively) cultured at physioxia.

Another mechanism of epigenetic regulation of gene expression involves a variety of non-coding RNA molecules, such as microRNA (miRNA), long non-coding RNA (lncRNA), and small nucleolar RNA (snoRNA). Johnston et al. measured the abundance of miRNA molecules in MCF-7 cells across a range of O_2_ pressures, from 18% O_2_ to 1% O_2_ [65]. Among the most strongly affected miRNAs, miR-675, miR-1293, miR-7974, miR-653, and miR-140 were found upregulated at 18% O_2_, compared to 4% O_2_ (physioxia). On the other hand, miR-758, miR-335, miR-1185-1, miR-1185-2, and miR-889 were upregulated at 4% O_2_, compared to 18% O_2_. These miRNA molecules have been implicated in processes such as apoptosis, proliferation, metastasis, and tumor suppression. In the study by Xie et al. where silencing of the *DLK1-DIO3* cluster was observed in hESCs grown at 18% O_2_ after 20 passages, decreased expression of several non-coding RNA molecules was shown, including the lncRNA *MEG3* and the snoRNA *SNORD114-3* [59]. The expression of these molecules was rescued in cells cultured at 5% O_2_. *MEG3* is known to inhibit tumor proliferation by interacting with p53. Accordingly, silencing of *MEG3* was correlated with reduced DNA-damage-induced apoptosis in hESCs and their differentiated hepatocyte-like cells grown at ambient O_2_ tension. Using microarray and bioinformatic analyses, Shi et al. identified 47 lncRNAs and 14 miRNAs differentially expressed in hDPSCs cultured at 18% O_2_ and 3% O_2_, revealing a complex gene network between the non-coding RNAs and mRNAs [66]. Further, functional enrichment analysis revealed biological processes, such as the HIF-1 pathway, the Wnt pathway, carbon metabolism, and glutathione metabolism, to be affected by O_2_, among other processes.

In summary, broad epigenetic responses to physiological O_2_ levels have been reported, with striking differences observed depending on the cell type. These responses include DNA methylation and hydroxylation, histone acetylation and methylation, and altered expression of non-coding RNAs. Future research should be directed to fully characterize the specific mechanisms governing these cell-type-specific responses.

## 5. Oxygen, Proliferation, and Senescence

The first observation made about the impact of culturing cells at O_2_ tensions above physiological conditions was a decrease in their proliferation rate. In 1958, Cooper et al. reported that lowering O_2_ tension below normoxia increased the growth rate of rabbit embryo kidney cells [67]. A later study found that lowering O_2_ from 18% to 10% O_2_ increases the replicative lifespan (i.e., Hayflick limit) of WI-38 fibroblasts [68]. Similar observations were made by Balin et al. [69]. Perhaps expectedly, immortalized cells are not as sensitive to the cytostatic effect of atmospheric O_2_, as Falanga and Kirsner showed that single-seeded human fibrosarcoma and immortalized 3T3 cells could proliferate without difficulty in standard O_2_ conditions (18% O_2_), in contrast to primary dermal fibroblasts [70]. Subsequent studies focused on the molecular mechanisms of the antiproliferative effects of atmospheric O_2_. Hyperoxia was established to be an inducer of replicative senescence, which is a process initiated by oxidative DNA damage and telomere shortening and mediated by p53- and p21-dependent cell cycle arrest [71].

Culture of MEFs at 18% O_2_ resulted in DNA mutations and senescence. G:C to T:A transversions, a signature mutation caused by DNA oxidation, were observed in hyperoxic cells. Senescence was not observed in MEFs grown at 3% O_2_ [72]. Similar observations were made by Parrinello et al., showing induction of senescence and DNA strand breaks in MEFs grown in 18% O_2_ [73]. Cardiac fibroblasts from adult murine ventricle cultured at 10% and 18% O_2_ exhibited greater ROS production and cell cycle arrest at G2/M, compared with cells at 3% O_2_ [13]. Induction of p21 and decreased telomerase activity were also detected in hyperoxia. Fibroblasts lacking p21 were resistant to the effects of higher O_2_ tension. Overexpression of p21 was confirmed at both mRNA and protein levels in muscle precursor cells (MPCs) exposed to 18% O_2_, compared to 5% O_2_, along with increased p53 transcriptional activity [74]. Estrada et al. reported that telomere shortening rate per division cycle is 24% higher in hMSCs cultured in normoxia, compared to in physioxia (here, 3% O_2_) [19]. Another study reported increased p38 phosphorylation, an upstream regulator of p21, along with concomitant p21 overexpression in hDPSCs cultured in 18% O_2_. Elevated HO-1 and NQO-1 protein levels were also detected, suggesting Nrf2 activation [15]. More recently, Bon-Mathier et al. demonstrated that primary neonatal mouse cardiomyocytes have a higher proliferative capacity when cultured at 3% O_2_, which was associated with a dedifferentiation of cardiomyocytes at physioxia [75].

Recent evidence has linked the Wnt/β-catenin signaling pathway to the mechanisms regulating the response to O_2_ at physioxia/hyperoxia. Stabilized β-catenin and upregulation of Wnt gene targets were observed in fetal mouse neural stem cells (NSCs) at physioxia, promoting proliferation and reducing spontaneous differentiation of NSCs [76]. These effects were sustained even in HIF-1α–deficient cells, indicating that the O_2_-sensitive mechanism of Wnt activation is independent of HIF. Further research is required to understand the involvement and mechanisms of Wnt in O_2_ sensing at physiological O_2_ tensions.

In conclusion, there is ample evidence showing that hyperoxic cell culture induces senescence in a variety of cells by promoting oxidative DNA damage and activating p53 and p21. Collectively, these results emphasize the importance of culturing primary cells at physioxia, in order for them to retain their proliferative capacity. More evidence regarding the link between O_2_ and cell differentiation is discussed in the next section.

## 6. Oxygen and Cell Differentiation

The transcriptional program controlling stemness and differentiation of embryonic stem cells is in part regulated by HIF. For example, the expression of OCT4, a critical transcription factor in mediating stemness and pluripotency of embryonic stem cells, is regulated by HIF-2α [77]. Westfall et al. showed that expression of known HIF-1/2α and OCT4 targets was markedly increased in human embryonic cell lines at 4% O_2_, compared to at 18% O_2_ [78]. The mechanisms governing hypoxia and stemness are reviewed in [79].

In accordance, lowering O_2_ to 5% increased the proliferation and differentiation of human macrophage progenitor cells [80,81]. Fehrer et al. found that while culturing hMSCs at 3% O_2_ extends their proliferative lifespan and inhibits senescence, their capacity of differentiating into osteoblasts and adipocytes was blunted at this O_2_ tension [82]. Similar mechanisms seem to affect osteoblast differentiation. Culture of bone-marrow-isolated adult multilineage-inducible cells at 3% O_2_ increased their proliferation but blocked their differentiation into osteoblasts, which is otherwise present when cultured at 18% O_2_ [83]. mRNA levels of *OCT4*, *HIF1A*, and *TERT* (telomerase reverse transcriptase) were upregulated at 3% O_2_. In contrast, lower O_2_ levels have been found to promote chondrogenesis, which is in agreement with the fact that cartilage is avascular and in hypoxic environment [84,85,86,87] (reviewed in [88]). Studer et al. reported that culture of mesencephalic precursor cells at 3% O_2_ enhances both their proliferation and differentiation into dopaminergic neurons [89]. Recent evidence has also shown that culture of stem cells at physioxia improves their therapeutic potential for clinical application [90,91]. The relevance of O_2_ levels for stem cell biology and their application in regenerative medicine is reviewed in detail in [92].

Immortalized cell lines are widely used in research as surrogates of primary cells, which are often less accessible than the former to the research community. A study found that the ability of the monocyte-derived cell line THP-1 to differentiate into macrophage-like-cells is affected by O_2_ tension. The metabolic activity and rate of drug-induced differentiation of THP-1 cells were enhanced at 5% O_2_, compared to 18%, with the primary macrophage-like functions of the differentiated cells being affected by O_2_ as well [93].

In summary, the O_2_ levels in culture have a profound impact on the differentiation capacity and stemness of cells, with specific effects varying among cell types, likely according to their oxygenation status in vivo.

## 7. Oxygen, Cell Bioenergetics, and Mitochondrial Dynamics

Since most cellular O_2_ is consumed by mitochondria during respiration and ROS production, it is unsurprising that broad oxygen-dependent effects on cellular metabolism and mitochondrial function been reported in a variety of cells. Estrada et al. showed that culturing hMSCs at 18% O_2_ enhances oxidative phosphorylation (OXPHOS) and decreases glycolysis, which, in turn, leads to higher rates of ROS production and oxidative-stress-related damage. In this study, they found a decreased oxygen consumption rate (OCR)/extracellular acidification rate (ECAR) ratio in hMSCs cultured at 3% O_2_, compared to at 18% O_2_, which was associated with increased expression of HIF-1/2 targets that encode enzymes involved in glucose metabolism, such as pyruvate dehydrogenase kinase, phosphofructokinase, and lactate dehydrogenase [19]. Enhanced activity of these enzymes increases the glycolytic rate and diverts pyruvate from entering the TCA cycle, while promoting its reduction to lactate in order to regenerate reduced nicotinamide dinucleotide (NAD^+^). In agreement with these results, Lees et al. found that hPSCs cultured in 18% O_2_ have an increased OCR and ATP production rate, compared to cells cultured at physioxia [60]. Although no significant differences were observed in basal ECAR, which is a proxy for glycolytic rate, RNA-seq revealed increased expression of genes related to glycolysis in cells grown at 5% O_2_. Using ^13^C glucose labeling, they also observed increased glycolytic intermediary metabolites (e.g., fructose-1,6-biphosphate) and lactate. In primary human corneal endothelial cells, no effect on the basal OCR was observed at 2.5% O_2_, compared to atmospheric O_2_ tension. In turn, basal ECAR was increased at 2.5% O_2_ [94]. This result is in accordance with the ability of the corneal endothelium to rely on glycolysis to meet the energy requirements of the cornea under lower oxygen availability. In another study, Zhu et al. demonstrated that culture of rat primary cortical neurons at physioxia improves mitochondrial function, as shown by increased mitochondrial membrane potential, mitochondrial dehydrogenase activity, and ATP levels at 5% O_2_, compared to at 18% O_2_ [21]. Although these outcomes suggest enhanced mitochondrial respiration, glucose uptake and consumption rates were also found to be higher at 5% O_2_. Further, Zhu et al. observed reduced glucose oxidation, measured by ^14^CO_2_ production, along with higher lactate levels in neurons cultured at 5% O_2_. AMPK phosphorylation was also found to be higher at physioxia. As such, many of the metabolic effects of culturing neurons at 5% O_2_ were abolished by pharmacological inhibition of AMPK.

A handful of other studies have reported opposite patterns, suggesting higher OXPHOS rates at physioxia. This could be explained by the fact that hyperoxia-induced ROS directly inactivate mitochondrial enzymes such as aconitase, pyruvate dehydrogenase complex, and respiratory complexes I and II (reviewed by Alva et al. in [95]). For example, MPCs isolated from old mice have a higher basal and maximal OCR at 3% O_2_, compared to 18% O_2_. In fact, when old MPCs are cultured at 3% O_2_, they have the same OCR as that of MPCs from young mice, which show no difference in OCR between the two O_2_ concentrations [25]. Old MPCs grown at hyperoxia seemed to compensate by upregulating glycolytic metabolism, as the ECAR measured was higher than that of old MPCs grown at physioxia and young MPCs at either O_2_ tension, which, again, did not seem to be affected much by O_2_. In addition to age, sex has also been shown to influence the bioenergetic response to different oxygen concentrations. Experiments with primary rat cortical astrocytes have uncovered sex differences in their bioenergetic profile when cultured at 3% O_2_. In contrast, no differences in bioenergetic parameters were found in astrocytes grown at 18% O_2_, denoting how physiological phenomena can be masked by culturing cells at nonphysiological oxygen tensions [96].

Studies with immortalized cell lines have shown similar trends. Timpano et al. studied the effects of O_2_ tension in the metabolism of four human cell lines, U87MG, HRPTEC, HEK293, and MCF-7. In general, cells had a lower mitochondrial metabolic activity when cultured at 12% O_2_ or higher, compared to at physioxia, as measured by fluorometric detection of resaruzin reduction [24]. Similarly, Moradi et al. investigated the effect of O_2_ concentration in cellular respiration using four human cancer cell lines. MCF-7, Huh-7 and LNCaP cells had a lower basal OCR at 18% O_2_, compared to at 5% O_2_, while no difference was observed in SaOS2 [97]. Maximal OCR was also lower at 18% O_2_ in all cell lines, except LNCaP, which had a greater maximal OCR at 18%. Collectively, these results indicate that physiological O_2_ levels seem to promote mitochondrial metabolism, although no effects on glycolytic rate (ECAR) were observed in this study.

The same study by Moradi et al. also revealed differential effects of O_2_ on mitochondrial network dynamics, which were analyzed using the MiNA tool [98]. Normoxia was associated with an increased mitochondrial footprint, indicative of mitochondrial abundance, in Huh-7 and SaOS2 cells, but had the opposite effect on LNCaP cells. In addition, the mean network size, which is indicative of the number of network branches, was lower in LNCaP and SaOS2 cells grown at 18% O_2_, while no significant difference was observed in MCF-7 and Huh-7. In the study by Timpano et al., a variety of effects in mitochondrial fusion and morphology of U87MG, HRPTEC, HEK293, and MCF-7 cells cultured in different O_2_ levels were measured. U87MG and HEK293 cells displayed a more globular mitochondrial morphology, when cultured at atmospheric O_2_, compared to cells cultured at physioxia, while HRPTEC cells showed no difference in mitochondrial morphology, and MCF-7 had a more globular mitochondrial morphology at 5% O_2_ than at 18% O_2_. In a different report, mitochondria in primary neurons were found to be longer and have a more elongated morphology at 5% and 2% O_2_ levels, compared to cells kept at atmospheric O_2_ [99]. Accordingly, mitochondrial network parameters, such as network size, mitochondrial perimeter, and mitochondrial fraction, were higher with decreasing O_2_ concentration.

In summary, O_2_ conditions affect mitochondrial respiration, glycolysis, mitochondrial morphology, and mitochondrial network dynamics (Table 1). Although, in some cases, the trend would seem to indicate that OXPHOS and highly fused mitochondrial networks are favored under physioxia, these are not widespread observations. This is likely due to the unique metabolic profiles of each type of cell, causing cells to react differently when challenged with varying O_2_ tensions. In any case, the effects of environmental O_2_ on metabolic and mitochondrial characteristics of cells can no longer be neglected and must be considered when conducting experiments that target the mitochondria and cellular metabolism.

## 8. Modeling Tissue Physiology and Pathology in Physioxia

One of the goals of cell culture is to mimic the behavior of cells and tissues in vivo, in both their physiological and pathological states. In this regard, physiological O_2_ conditions have been shown to facilitate the maintenance of in vivo-like properties of cells cultured in vitro. For instance, when compared to cells grown in physioxia, microvascular endothelial cells isolated from the foreskins of neonates cultured in normoxia exhibit a differential expression of proteins related to the functionality of endothelial cells, such as type IV collagen, platelet endothelial cell adhesion molecule-1 (PECAM-1), and von Willebrand factor. In addition, vascular channel formation induced by collagen gels was increased in 18% O_2_ [100]. Piossek et al. showed that co-culture of human renal proximal tubular epithelial cells with human fibroblasts at physioxia improves their functional properties, as measured by epithelial barrier integrity, expression of membrane transporters, and transport of cations and anions [101]. Another study showed that hyperoxia promotes hepatocyte dedifferentiation in vitro, confirmed by the observation that mouse hepatocytes cultured in normoxia (18% O_2_) undergo epithelial-to-mesenchymal transition, obtain fibroblast-like morphology, and show impaired hepatic functions. These effects were prevented by culturing hepatocytes at 5% O_2_, which show increased glycogen stores, increased LDL-uptake ability, and sustained expression of cytochrome P450 enzymes, all important characteristics of hepatocytes in vivo [102].

Intracellular Ca^2+^ concentration ([Ca^2+^]_i_) is critical in mediating a wide variety of processes, in both physiological and pathological conditions. Keeley et al. have recently shown that culturing endothelial cells at atmospheric O_2_ levels impairs Ca^2+^ handling. Using human umbilical vascular endothelial cells (HUVECs) cultured at 5% O_2_, they reported a negative feedback mechanism mediated by protein phosphatase 2A that regulates Ca^2+^–dependent NO biosynthesis in response to histamine, which is not observed at atmospheric O_2_ [103]. Interestingly, although they observed a similar intracellular Ca^2+^ peak in response to histamine at both O_2_ conditions, [Ca^2+^]_i_ at the plateau phase was reduced in cells cultured in 5% O_2_, suggesting enhanced cytosolic Ca^2+^ removal mechanisms. Similar results were obtained when HUVECs were treated with ATP and ionomycin. These observations were further characterized in a subsequent study by the same lab. The difference in plateau [Ca^2+^]_i_ upon stimulation with histamine at 5% and 18% O_2_ was absent when cells were treated with cyclopiazonic acid, an inhibitor of the sarco/endoplasmic reticulum Ca^2+^–ATPase (SERCA). Moreover, agonist-stimulated phosphorylation of phospholamban, an endogenous negative regulator of SERCA, was higher at 5% O_2_ than at ambient air. Consequently, enhanced SERCA activity at physioxia blunted ionomycin-induced Ca^2+^ overload and cell death, compared to the 18% O_2_ group [104]. Taken together, these results suggest that hyperoxic conditions in cell culture dysregulate Ca^2+^ handling by reducing SERCA activity, which may artificially increase the sensitivity of cells to cytotoxic stimuli that target intracellular Ca^2+^. Regulation of additional cytosolic Ca^2+^ influx and efflux mechanisms at 5% and 18% O_2_ should be investigated.

Hypoxia/reoxygenation (H/R) is the in vitro model of ischemia/reperfusion injury that takes place in pathologies such as strokes and myocardial infarctions. It involves exposing cells to hypoxia (≤ 1% O_2_) for a sustained period, usually coupled with nutrient deprivation, followed by placing them back to normoxic conditions. Researchers have hypothesized that oxidative stress and cell injury are artefactually exacerbated at the relatively hyperoxic reoxygenation conditions (at least an 18-fold pO_2_ increase, compared to hypoxia) used in most studies and that such stress may be less severe at physioxia (~5-fold pO_2_ increase, compared to hypoxia). Warpisnki et al. reported that H/R did not seem to increase the generation of mitochondrial and extramitochondrial O_2_^–^ at physioxia. Consequently, no protective effects of sulforaphane pretreatment were observed at 5% O_2_, compared to at ambient air [105]. Similarly, Danilov and Fiskum cultured rat cortical astrocytes at 7% or 18% O_2_ and exposed them to both hypoxia and glucose deprivation and then to reoxygenation in their respective O_2_ tensions [106]. Decreased cell death and lower levels of protein nitration and DNA oxidation were detected during reoxygenation at 7% O_2_, again confirming that hyperoxic reoxygenation under ambient air exacerbates cell injury in H/R models.

In summary, keeping physiological O_2_ conditions in cell culture will provide us a better understanding of the physiological and pathological processes of cells and, in turn, allow us to conduct more reliable and relevant experiments to study the effects of different stimuli on cell physiology in vitro. In the next section, we will discuss how the cellular response to drugs, hormones, and toxicants is dependent on O_2_ tension, leading to artifacts caused by supraphysiological O_2_ levels.

## 9. Cellular Response to Drugs, Hormones, and Toxicants

In vitro studies have provided us invaluable knowledge about the cellular effects and mechanisms of endogenous and exogenous substances, such as drugs, hormones, pollutants, toxins, and nanoparticles. However, experimental findings from cell culture often translate poorly in vivo. Recent evidence has shown that culturing cells in supraphysiological O_2_ levels can lead to an overestimation or underestimation of the potency of biologically active compounds or even demonstrate opposing effects. For example, resveratrol acts as a prooxidant in C2C12 and PC-3 cells grown in low glucose DMEM at 18% O_2_, as shown by an increased H_2_O_2_ efflux rate. However, this effect is absent in both cell lines when cultured at 5% O_2_ [107]. Yan et al. reported that the hepatotoxic effect of acetaminophen was less pronounced at physioxia, compared to atmospheric O_2_ [36]. On the other hand, the cytotoxic effects of copper oxide nanoparticles (CuO NPs) were diminished in normoxia, likely due to a preconditioning to oxidant insults in the hyperoxic environment of standard cell culture [108]. Similarly, the antiproliferative and cytotoxic effect of teriflunomide on the colon cancer cell lines SW480 and SW620 was significantly enhanced with decreasing O_2_ concentration [109]. Table 2 summarizes the findings in the literature on biologically active substances that have been shown to elicit different cellular responses at physioxia versus standard O_2_ conditions. Indeed, these observations highlight the need for utilizing physiologically representative O_2_ levels in cell culture, to improve the predictive value of in vitro pharmacological and toxicological studies.

## 10. Conclusions and Future Directions

An increasing amount of evidence shows that culturing cells in the hyperoxic environment of ambient air disturbs their redox homeostasis, alters gene expression, inhibits proliferation, impairs differentiation, impacts energy metabolism, and alters the cellular response to drugs (illustrated in Figure 2). The potential artifacts caused by growing cells at nonphysiological O_2_ levels underline the need for revisiting previous results from cell-culture-based research, particularly the ones intimately related with oxygen and oxidative stress, including, but not limited to, studies using hypoxia and hyperoxia models.

Additional research should be conducted to further characterize the mechanisms of altered cellular responses to O_2_ at 18% O_2_, compared to at physioxia, with a focus on elucidating the molecular basis of the response in this range and its specificity between different cell types. In particular, future research on oxidative post-translational modifications through the use of redox proteomics, in conjunction with other “omics” approaches, will be important. In addition, more research should be conducted to delineate the role of HIF at physiological O_2_ levels. While HIF stabilization/activation reported by some studies could well be the result of an acute reduction in O_2_ tension, the detection of HIF targets in a handful of cell types adapted to physioxia for at least two weeks [37,50,118] suggests a role for HIF activity at physioxia. In this respect, it is also critical to keep in mind that physioxia is a range rather than a set value, as it varies depending on the tissue. For example, while 3–6% O_2_ would be appropriate for cells like hepatocytes, cardiomyocytes, neurons, and enterocytes, the same would not be true for alveolar epithelial cells, which are exposed to ~13% O_2_ in vivo. On the other hand, most tumors exhibit median O_2_ levels below 2% [3]. As such, O_2_ tensions relevant to the specific tissue being studied should be utilized. Moreover, it is important to consider potential differences in pericellular O_2_ levels (in the liquid phase) versus the incubator headspace O_2_ (gaseous phase). Pericellular O_2_ levels depend on many factors, such as the O_2_ consumption rate of each cell type, the cell seeding density, the material of the cell culture vessel, and the volume of the media in the vessel, to mention a few [119]. These and other considerations are elegantly discussed in the seminal review by Keeley and Mann [2].

The findings discussed in this review highlight the need to incorporate incubators capable of regulating O_2_ concentration into our research facilities. While the elevated cost of commercially available O_2_-regulating incubators (>USD 10,000) may seem like a barrier to implement physioxia in cell culture workflows, our lab has recently developed a portable, home-made incubator with a total production cost of <USD 1000, which effectively regulates O_2_, CO_2_, and temperature [120]. We also demonstrated its ability to maintain both the headspace and pericellular levels at 5% O_2_ for several weeks. Similarly, Marchus et al. developed an O_2_-regulating incubator with a cost of ~USD 800 [121], showing that implementing physiological O_2_ conditions into routine cell culture does not need to come at a great expense.

Finally, additional considerations should be made to recreate a more physiologically representative environment in cell culture, such as using physiological culture media (e.g., Plasmax, HPLM), 3D culture, and co-culture (see [1]). Implementing physiological cell culture conditions will improve the quality and validity of studies conducted in vitro.

## Figures and Tables

**Figure 1 cells-11-03123-f001:**
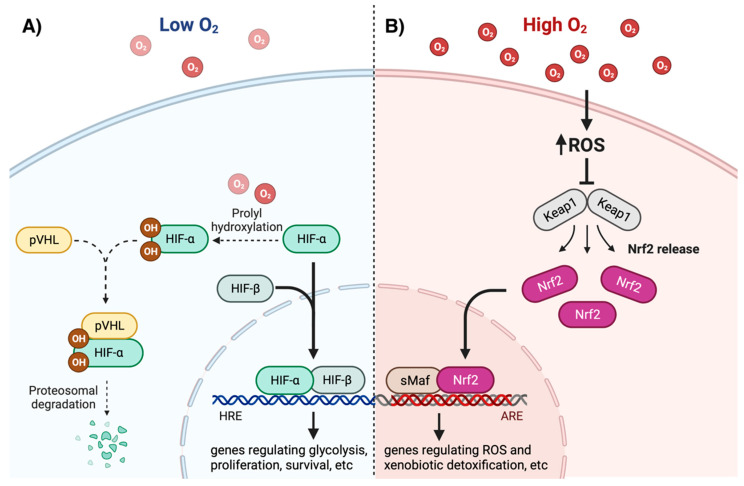
Pathways for intracellular oxygen/redox status sensing by HIF-1/2 and Nrf2. (**A**) Low O_2_ levels reduce the rate of HIF-1/2α hydroxylation, which leads to molecular recognition by the Von Hippel-Lindau tumor suppressor protein (pVHL) and subsequent proteasomal degradation. Stabilized HIF-1/2α binds HIF-β and translocates to the nucleus, where it binds the hypoxia response element (HRE) in DNA, inducing expression of glycolytic, proliferative, and pro-survival genes, among others. (**B**) High O_2_ levels promote increased reactive oxygen species (ROS) production. ROS-mediated oxidation of Keap1 releases Nrf2 preventing its proteasomal degradation. Nrf2 then translocates to the nucleus, heterodimerizes with small musculo-aponeurotic fibrosarcoma (sMaf), and binds to the antioxidant response element (ARE), inducing expression of genes involved in ROS and xenobiotic detoxification. Created with BioRender.com (accessed on 14 July 2022).

**Figure 2 cells-11-03123-f002:**
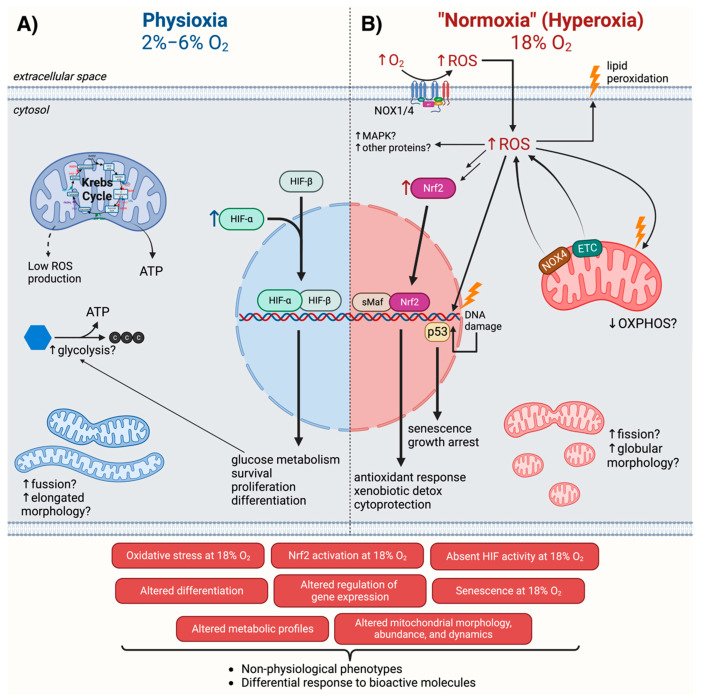
Cellular mechanisms differentially regulated at physioxia (2–6% O_2_) versus normoxia (18% O_2_). In physioxia (**A**), activation of HIF may drive the induction of genes that regulate glucose metabolism, survival, proliferation, differentiation, and other processes. In many cases, mitochondria have a more elongated morphology and form highly fused networks. In normoxia (**B**), increased ROS production by NADPH oxidases (NOX) and the electron transport chain (ETC) leads to oxidative stress, characterized by DNA and lipid oxidation, and promotes activation of a variety of pathways such as p53 and Nrf2, which, in turn, results in the induction of antioxidant genes and senescence. Further, in many instances, mitochondria show a more globular morphology and increased fission of mitochondrial networks. These cellular mechanisms result in nonphysiological phenotypes and differential responses to bioactive molecules. Created with BioRender.com (accessed on 26 September 2022).

**Table 1 cells-11-03123-t001:** Effects of O_2_ tension (between 2–18% O_2_) on cellular metabolism, mitochondrial abundance, morphology, and network dynamics.

Cell Types	Experimental Conditions	Methods	Outcomes	Reference
*Metabolic Effects*				
Stem cells	hMSCs/18% or 3% O_2_	Seahorse XF analysis, microarray, RT-PCR	↑ OCR/ECAR ratio at 18% O_2_ ↑ expression HIF targets involved in glucose metabolism at 3% O_2_	[19]
	hPSCs/18% or 5% O_2_	Seahorse XF analysis, NMR spectroscopy, RNA-seq, RT-PCR	↑ OCR/ECAR ratio at 18% O_2_ ↑ glycolytic intermediates at 3% O_2_ ↑ expression HIF targets involved in glucose metabolism at 3% O_2_	[60]
	MPCs from old mice/18% or 3% O_2_	Seahorse XF analysis	↓ OCR at 18% O_2_ ↑ ECAR at 18% O_2_	[25]
Primary differentiated cells	hCEnCs/18% or 2.5% O_2_	Seahorse XF analysis	↑ ECAR at 2.5% O_2_	[94]
	Rat primary cortical neurons/18% or 5% O_2_	ATP bioluminescent assay, LSC, lactate assay	↑ glucose uptake at 5% O_2_ ↓ glucose oxidation at 5% O_2_ ↑ lactate levels at 5% O_2_	[21]
	HRPTEC/18–3% O_2_	Resazurin assay	↑ metabolic activity at 18% O_2_, compared to 15% and 12% O_2_	[24]
Cancer/immortalized cells	U87MG/18–3% O_2_	Resazurin assay	↓ metabolic activity at 18% O_2_, compared to 8–3% O_2_	
	MCF-7/18–3% O_2_	Resazurin assay	↓ metabolic activity at 18% O_2_, compared to 8% O_2_	
	MCF-7/18% or 5% O_2_	Seahorse XF analysis	↓ basal and maximal OCR at 18% O_2_	[97]
	LNCaP/18% or 5% O_2_	Seahorse XF analysis	↓ basal OCR at 18% O_2_ ↑ maximal OCR at 18% O_2_	
	Huh-7/18% or 5% O_2_	Seahorse XF analysis	↓ basal and maximal OCR at 18% O_2_	
	SaOS2/18% or 5% O_2_	Seahorse XF analysis	↓ maximal OCR at 18% O_2_	
*Effects on mitochondrial morphology, abundance, and dynamics*				
Primary differentiated cells	rat primary neurons/18%, 5%, or 2% O_2_	TEM and confocal microscopy; Image J	Globular-shaped mitochondria at 18% O_2_ (versus elongated at 2% and 5% O_2_) ↓ mitochondrial network size, mitochondrial fraction, and mitochondrial perimeter at 18% O_2_	[99]
Cancer/immortalized cells	U87MG/18–3% O_2_	Confocal microscopy; Volocity	Rounder mitochondria at 18% O_2_	[24]
	HEK293/18–3% O_2_	Confocal microscopy; Volocity	Rounder mitochondria at 18% O_2_	
	MCF-7/18–3% O_2_	Confocal microscopy; Volocity	Elongated mitochondria at 18% O_2_	
	LNCaP/18% or 5% O_2_	Confocal microscopy; MiNA	↓ mitochondrial footprint at 18% O_2_ ↓ mean network size 18% O_2_	[97]
	Huh-7/18% or 5% O_2_	Confocal microscopy; MiNA	↑ mitochondrial footprint at 18% O_2_	
	SaOS2/18% or 5% O_2_	Confocal microscopy; MiNA	↑ mitochondrial footprint at 18% O_2_ ↓ mean network size 18% O_2_	

Abbreviations: ECAR, extracellular acidification rate; hCEnCs, human corneal endothelial cells; hMSCs, human mesenchymal stem cells; hPSC, human pluripotent stem cells; HRPTEC, human renal proximal tubule epithelial cells; LSC, liquid scintillation counting; MiNA, mitochondrial network analysis; NMR, nuclear magnetic resonance; OCR, oxygen consumption rate; TEM, transmission electron microscopy; XF, extracellular flux.

**Table 2 cells-11-03123-t002:** Differential cellular responses to drugs, hormones, and toxicants at atmospheric O_2_ versus physioxia.

Molecule	Mechanism	Conditions	Outcomes	Reference
*Drugs*				
resveratrol	ROS scavenger, multiple targets	PC-3 and C2C12 cells 18% or 5% O_2_	Differential H_2_O_2_ production, proliferation, and mitochondrial network dynamics	[107]
sulforaphane	ROS scavenger, multiple targets	bEnd.3 cells 18% or 5% O_2_ H/R	Attenuated reoxygenation-induced ROS production at 18% O_2_ but not at 5% O_2_	[105]
quercetin	ROS scavenger, multiple targets	human neonatal foreskin fibroblasts 18% or 4% O_2_	GSH depletion and loss of type I cells at 18% O_2_ but not at 4% O_2_	[110]
doxorubicin	DNA intercalating agent	HCT116, IMR90, U2OS, and MCF-7 cells 18% O_2_ or 5% O_2_	↑ apoptosis at 18% O_2_	[49]
acetaminophen	COX inhibitor	mouse hepatocytes 18%, 10%, or 5% O_2_	↑ hepatotoxicity at 18% O_2_ ↑ mROS and RNS production at 18% O_2_	[36]
		HepG2 cells 18%, 8%, or 3% O_2_	↓ hepatotoxicity at 18% O_2_ differential regulation of phase I and II enzymes	[111]
cyclophosphamide	DNA cross-linking agent	HepG2 cells 18%, 8%, or 3% O_2_	↓ hepatotoxicity at 18% O_2_	[111]
teriflunomide	pyrimidine synthesis inhibitor	SW480 and SW620 cells 18% or 10% O_2_	↓ proapoptotic effect at 18% O_2_ ↓ antiproliferative effect at 18% O_2_	[109]
oxaliplatin	DNA synthesis inhibitor	SW480 and SW620 cells 18% or 10% O_2_	↓ antiproliferative effect at 18% O_2_	[109]
paclitaxel	microtubule stabilizer	mouse mammary tumors 18% or 3–5% O_2_	↑ cytotoxicity at 18% O_2_	[47]
alpelisib	PI3K inhibitor	mouse mammary tumors 18% or 3–5% O_2_	↑ cytotoxicity at 18% O_2_	[47]
erlotinib	EGFR inhibitor	mouse mammary tumors 18% or 3–5% O_2_	↑ cytotoxicity at 18% O_2_	[47]
vemurafenib	BRAF^V600^ inhibitor	patient-derived melanoma cells 18% or 6% O_2_	↓ Ki-67-positive cells at 18% O_2_ ↓ reduction of VEGF, PCG-1α, and SLC7A11 levels at 18% O_2_	[37]
trametinib	MEK1/2 inhibitor	patient-derived melanoma cells 18% or 6% O_2_	↓ Ki-67-positive cells at 18% O_2_ ↓ reduction of VEGF, PCG-1α, and SLC7A11 levels at 18% O_2_	[37]
camptothecin	topoisomerase inhibitor	U87MG cells 18% O_2_ or 9% O_2_	↑ cytotoxicity at 18% O_2_	[112]
dimethyl fumarate	Nrf2 inducer	RAW 264.7 cells 18% O_2_ or 5% O_2_	↑ expression of Nrf2 targets and antioxidant response	[20]
glycolic acid	keratolytic, antioxidant	Hs68 and HaCaT cells 18% or 2% O_2_	Differential regulation of skin barrier and dermal network-related genes	[113]
gluconolactone	keratolytic, antioxidant	Hs68 and HaCaT cells 18% or 2% O_2_	Differential regulation of skin barrier and dermal network-related genes	[113]
salicylic acid	keratolytic, AMPKactivator	Hs68 and HaCaT cells 18% or 2% O_2_	Differential regulation of skin barrier and dermal network-related genes	[113]
*Hormones*				
17β-estradiol	ER antagonist	C2C12 cells 18% O_2_ or 5% O_2_	Differential H_2_O_2_ production, metabolism, and mitochondrial network dynamics	[114]
*Toxicants*				
LPS	TLR4 agonist	RAW 264.7 cells 18% O_2_ or 5% O_2_	↑ production of inflammatory mediators	[20]
rotenone	complex I inhibitor	SH-SY5Y cells 18% O_2_ or 5% O_2_	↓ cytotoxicity at 18% O_2_ No inhibition of ATP synthesis with 0.2 µM rotenone at 18% O_2_ (with effects observed at 5%)	[115]
acrolein	DNA and protein adduct inducer	differentiated H9c2 cells 18% O_2_ and 5% O_2_	↑ cytotoxicity at 18% O_2_	[116]
aflatoxin B	DNA adduct inducer	HepG2 cells 18%, 8%, or 3% O_2_	↓ hepatotoxicity at 18% O_2_	[111]
*Other*				
*V. baccifera* leaf extract	Prooxidant, cytotoxic	HepG2 cells 18% O_2_ or 8% O_2_	↑ cytotoxicity at 18% O_2_	[117]
CuO NPs	Prooxidant, genotoxic, cytotoxic	A549 cells 18% O_2_ or 13% O_2_	↓ NP-induced oxidative stress at 18% O_2_ ↓ cytotoxicity at 18% O_2_	[108]

Abbreviations: AMPK, AMP-dependent kinase; BRAF^V600^, B-raf (mutated); COX, cyclooxygenase; EGFR, epidermal growth factor receptor; ER, estrogen receptor; GSH, reduced glutathione; H/R, hypoxia/reoxygenation; LPS, lipopolysaccharide; MEK1/2, MAPK/ERK kinase 1/2; NPs, nanoparticles; PI3K, phosphatidylinositol 3-kinase; ROS, reactive oxygen species; mROS, mitochondrial ROS; RNS, reactive nitrogen species, TLR4, toll-like receptor 4.

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
