# Peer review of "Supraphysiological Oxygen Levels in Mammalian Cell Culture: Current State and Future Perspectives"

_cells, 2022, doi:10.3390/cells11193123_

Round 1
Reviewer 1 Report
This review provides a comprehensive and thoughtful update on the importance of conducting cell culture experiments under oxygen levels encountered by tissue/cells in vivo. The review certainly complements and extends previous reviews and carefully discusses recent primary papers.
Specific comments:
1. The authors may consider further updating their review by citing the following additional recent publications in the field (see ) and the recent July 2022 Expert Recommendation by Sies et al. in Nature Rev Mol Cell Biol. 23: 499-515, 2022.
2. For the field is critically important that the authors to carefully 're-define' the terms to denote atmospheric (notably hyperoxic), physioxic (tissue/cell specific normoxia) and hypoxic (usually < 1% oxygen) O2 levels. There are sentences/text where the authors are discuss published findings and use the term 'normoxia' (as used incorrectly in the publications) to denote atmospheric oxygen levels. The authors should clearly state that equating atmospheric, standard cell culture oxygen levels with 'normoxia' is incorrect. That said, the authors have elegantly discussed research from multiple groups engaged in Physioxia research.
2. Fig. 1, lines 140-151: The description of NRF2 regulated antioxidant gene/protein expression in oxidative stress requires minor rewriting so that readers are informed of the current understanding of KEAP1/NRF2 signaling and nuclear translocation and ARE gene activation. I suggest the authors cite the following state-of-the-art reviews by established experts (e.g. Masayuki Yamamoto, John Hayes, Thomas Kensler) in this field rather that guiding reader initially to Ref 36. (see Susuki & Yamamoto, FRBM 58: 93-100, 2015; Physiol Rev. 2018 Jul 1;98(3):1169-1203)
3. Relevant earlier study in macrophages would complement the review (see Haas et al., J. Cell Physiol. 230:1128-38, 2015) and recent study highlighting effects of physioxia on nitric oxide bioavailability and sensitivity of cells to medium iron supplementation (see Sevimli et al., Redox Biol. 2022 Jul;53:102319. doi: 10.1016/j.redox.2022.102319)
4. Conclusions/Future Directions - This section is written rather succinctly and the authors primarily address the difficulty of setting up oxygen regulated workstations. Their economical devices are certainly important for researchers in the field.
I feel the authors have not sufficiently addressed the importance of establishing a 'physiological oxygen phenotype' in vitro. I am referring to the stabilisation HIF and downstream HIF target gene expression following acute (hours and 1-2 days) oxygen reduction. To obviate confounding effects of HIF, cells deemed to be cultured under physioxia oxygen levels (true normoxia is tissue/organ specific), cells require longer-term adaptation to a defined oxygen levels. Guidelines have been published by Ferguson et al. (2018), Keeley & Mann (2019), and recently by Sies et al. (2022) and I recommend the authors consider these.
Author Response
Reviewer #1
This review provides a comprehensive and thoughtful update on the importance of conducting cell culture experiments under oxygen levels encountered by tissue/cells in vivo. The review certainly complements and extends previous reviews and carefully discusses recent primary papers.
Specific comments:
- The authors may consider further updating their review by citing the following additional recent publications in the field (see ) and the recent July 2022 Expert Recommendation by Sies et al. in Nature Rev Mol Cell Biol. 23: 499-515, 2022.
Response: We have added this citation to section 2. - For the field is critically important that the authors to carefully 're-define' the terms to denote atmospheric (notably hyperoxic), physioxic (tissue/cell specific normoxia) and hypoxic (usually < 1% oxygen) O2 levels. There are sentences/text where the authors are discuss published findings and use the term 'normoxia' (as used incorrectly in the publications) to denote atmospheric oxygen levels. The authors should clearly state that equating atmospheric, standard cell culture oxygen levels with 'normoxia' is incorrect. That said, the authors have elegantly discussed research from multiple groups engaged in Physioxia research.
Response: This is a very good comment and definitely a matter of debate between the authors while writing the manuscript. We are aware that “normoxia” is incorrectly used to refer to near atmospheric O2 levels. However, although some authors have chosen to redefine it and apply this term for physiological O2 tension, we feel it is more complicated for the scientific community to change the definition of a term that is used universally. Rather, we chose to apply the newer term “physioxia” to refer to physiological O2 tensions, and keep “normoxia” to refer to near atmospheric O2, while clearly stating that this “normoxia” is in fact hyperoxic relative to in vivo conditions. For clarity, we have explained the use of these terms throughout the manuscript in the introduction “In this review, we will use the term ‘physioxia’ (first used in [3]) to describe O2 levels in this 2%–6% range, and ‘normoxia’ for the hyperoxic (18%) conditions of standard cell culture.”. - Fig. 1, lines 140-151: The description of NRF2 regulated antioxidant gene/protein expression in oxidative stress requires minor rewriting so that readers are informed of the current understanding of KEAP1/NRF2 signaling and nuclear translocation and ARE gene activation. I suggest the authors cite the following state-of-the-art reviews by established experts (e.g. Masayuki Yamamoto, John Hayes, Thomas Kensler) in this field rather that guiding reader initially to Ref 36. (see Susuki & Yamamoto, FRBM 58: 93-100, 2015; Yamamoto et al., Physiol Rev. 2018 Jul 1;98(3):1169-1203)
Response: We have rephrased that paragraph and expanded our description of the Nrf2 pathway, citing the following refs:
Suzuki and Yamamoto 2015; doi: 10.1016/j.freeradbiomed.2015.06.006
Yamamoto et al. 2018; doi: 10.1152/physrev.00023
Suzuki et al. 2019; doi: 10.1016/j.celrep.2019.06.047
In addition, we have cited McMahon et al. 2010 and Chen et al. 2005 to exemplify how electrophilic lipid peroxidation end-products (e.g., 4-HNE) can induce Nrf2 activation.
- Relevant earlier study in macrophages would complement the review (see Haas et al., J. Cell Physiol. 230:1128-38, 2015) and recent study highlighting effects of physioxia on nitric oxide bioavailability and sensitivity of cells to medium iron supplementation (see Sevimli et al., Redox Biol. 2022 Jul;53:102319. doi: 10.1016/j.redox.2022.102319)
Reponse: Thank you for pointing us to the article by Haas et al. We have cited this article in the second paragraph of section 2, where we state that increased ROS production at supraphysiological O2 tensions is observed in a variety of cell types. We have also included this reference in Table 1. Although the study by Sevimli et al. provides an important breakthrough, we do not consider that it would fit properly in any of the current sections of the manuscript without disrupting its flow, since it would seem like an isolated idea or concept that is not necessarily connected with the rest. - Conclusions/Future Directions - This section is written rather succinctly and the authors primarily address the difficulty of setting up oxygen regulated workstations. Their economical devices are certainly important for researchers in the field.
I feel the authors have not sufficiently addressed the importance of establishing a 'physiological oxygen phenotype' in vitro. I am referring to the stabilisation HIF and downstream HIF target gene expression following acute (hours and 1-2 days) oxygen reduction. To obviate confounding effects of HIF, cells deemed to be cultured under physioxia oxygen levels (true normoxia is tissue/organ specific), cells require longer-term adaptation to a defined oxygen levels. Guidelines have been published by Ferguson et al. (2018), Keeley & Mann (2019), and recently by Sies et al. (2022) and I recommend the authors consider these.
Response: Thank you for these highly valuable suggestions. We have expanded the conclusion to further discuss the technical challenges and recommendations for implementing physioxia in cell culture workflows. Regarding the confounding effects of HIF following acute reduction of O2 tension, several studies have shown a sustained increase in HIF targets in cells adapted to physioxia for at least 2 weeks (Alva et al. 2022; Osrodek et al. 2021), suggesting a role for HIF activity at physioxia rather than it being a mere consequence of an acute reduction of O2 tension while performing the experiments. We have added an extra sentence in the 2nd paragraph of section 3 to discuss this. We have also added a couple sentences to the conclusion to state the need for further delineating the possible role of HIF at physiological oxygen levels.
Furthermore, we have written a new paragraph delineating some technical considerations that need to be made when implementing physioxia in cell culture, citing the seminal review by Keeley and Mann (2019).
In addition to the changes requested, we have performed minor editorial changes to improve the quality of the manuscript.
Reviewer 2 Report
This review article is a comprehensive summary of the effects of different oxygen concentrations during cell culture. Cell culture under atmospheric conditions has been analyzed under conditions that deviate from the partial pressure of oxygen in vivo, and there has been controversy over the years as to whether it reproduces the physiological cellular environment. Efforts continue to be made to bring the extracellular environment closer to that in vivo, including the type and concentration of extracellular matrix, three-dimensional culture conditions, and mechanical stimuli. Against this background, I believe that this review article is highly important and deserves to be published because of its focus on oxygen concentrations. However, I would like to see improvement on the following comments.
1. Oxygen concentration conditions should be unified as either “phisioxia” or “normoxia”.
2. A summary table should be added at the end of each section to facilitate readers' understanding.
3. The conclusion section is ambiguous and should be rewritten or added.
Author Response
Reviewer #2
This review article is a comprehensive summary of the effects of different oxygen concentrations during cell culture. Cell culture under atmospheric conditions has been analyzed under conditions that deviate from the partial pressure of oxygen in vivo, and there has been controversy over the years as to whether it reproduces the physiological cellular environment. Efforts continue to be made to bring the extracellular environment closer to that in vivo, including the type and concentration of extracellular matrix, three-dimensional culture conditions, and mechanical stimuli. Against this background, I believe that this review article is highly important and deserves to be published because of its focus on oxygen concentrations. However, I would like to see improvement on the following comments.
- Oxygen concentration conditions should be unified as either “phisioxia” or “normoxia”.
Response: Although we recognize that it may seem repetitive, we believe that it is useful for readers to be reminded of the O2tension(s) that “physioxia” and “normoxia” represent, mainly because 1) these concepts may be new for some readers that are used to not being concerned about O2 levels in vitro and 2) physioxia is not a set value but rather a range that is tissue-specific and thus it’s important to state the concentration used in each study.
- A summary table should be added at the end of each section to facilitate readers' understanding.
Response: We feel that a having a summarizing table at the end of each section could negatively impact the manuscript because it would make it unnecessarily longer and impact the reader’s experience. However, we have added a table at the end of the section “Oxygen, Cell Bioenergetics, and Mitochondrial Dynamics”. Due to its content, with opposing trends and very cell-type specific effects on cellular metabolism, a table to summarize all of the findings would be beneficial. In addition, we have added a new figure (figure 2) in the conclusions section to summarize the mechanisms described in this review and facilitate the reader’s understanding of this topic.
- The conclusion section is ambiguous and should be rewritten or added.
Response: We have modified the conclusion to: 1) make the key takeaways clearer, 2) discuss some technical considerations that must be made when implementing physioxia, and 3) expand on future directions.
In addition to the changes requested, we have performed minor editorial changes to improve the quality of the manuscript.